# Small Bowel Perforation Due to Renal Carcinoma Metastasis: A Comprehensive Case Study and Literature Review

**DOI:** 10.3390/diagnostics14070761

**Published:** 2024-04-03

**Authors:** Đorđe Todorovic, Bojan Stojanovic, Milutinovic Filip, Đorđe Đorđevic, Milos Stankovic, Ivan Jovanovic, Marko Spasic, Bojan Milosevic, Aleksandar Cvetkovic, Dragce Radovanovic, Marina Jovanovic, Bojana S. Stojanovic, Damnjan Pantic, Danijela Cvetkovic, Dalibor Jovanovic, Vladan Markovic, Milica Dimitrijevic Stojanovic

**Affiliations:** 1Department of Urology, University Clinical Center Kragujevac, 34000 Kragujevac, Serbia; djtodorovic87kg@gmail.com (Đ.T.); filipmilutinovic582@gmail.com (M.F.); drdjoledjordjevic95@gmail.com (Đ.Đ.); damnjanpantic@yahoo.com (D.P.); 2Department of Surgery, Faculty of Medical Sciences, University of Kragujevac, 34000 Kragujevac, Serbia; milos_st_kg@yahoo.com (M.S.); drbojanzm@gmail.com (B.M.); draleksandarcvetkovic@gmail.com (A.C.); drakce_5@hotmail.com (D.R.); 3Center for Molecular Medicine and Stem Cell Research, Faculty of Medical Sciences, University of Kragujevac, 34000 Kragujevac, Serbia; ivanjovanovic77@gmail.com; 4Department of Internal Medicine, Faculty of Medical Sciences, University of Kragujevac, 34000 Kragujevac, Serbia; marinna034@gmail.com; 5Department of Pathophysiology, Faculty of Medical Sciences, University of Kragujevac, 34000 Kragujevac, Serbia; bojana.stojanovic04@gmail.com; 6Department of Genetics, Faculty of Medical Sciences, University of Kragujevac, 34000 Kragujevac, Serbia; danijelac7@gmail.com; 7Department of Pathology, Faculty of Medical Sciences, University of Kragujevac, 34000 Kragujevac, Serbia; dalekg84@gmail.com (D.J.); milicadimitrijevic@yahoo.com (M.D.S.); 8Department of Radiology, Faculty of Medical Sciences, University of Kragujevac, 34000 Kragujevac, Serbia; drjack.vm@gmail.com

**Keywords:** renal cell carcinoma, small bowel perforation, metastasis, acute abdomen

## Abstract

This case report presents a unique instance of small bowel perforation caused by solitary metastasis from renal cell carcinoma (RCC), a rare and complex clinical scenario. The patient, a 59-year-old male with a history of RCC treated with nephrectomy four years prior, presented with acute abdomen symptoms. Emergency diagnostic procedures identified a significant lesion in the small intestine. Surgical intervention revealed a perforated jejunal segment due to metastatic RCC. Postoperatively, the patient developed complications, including pneumonia and multi-organ failure, leading to death 10 days after surgery. Histopathological analysis confirmed the metastatic nature of the lesion. This case underscores the unpredictable nature of RCC metastasis and highlights the need for vigilance in post-nephrectomy patients. The rarity of small bowel involvement by RCC metastasis, particularly presenting as perforation, makes this case a significant contribution to medical literature, emphasizing the challenges in the diagnosis and management of such atypical presentations.

## 1. Introduction

Renal cell carcinoma (RCC) is a predominant form of kidney cancer, known for its unpredictable metastatic behavior and its ability to spread to various organs [1]. While the common sites for RCC metastasis include the lungs, bones, and brain, metastatic involvement of the gastrointestinal tract, particularly the small intestine, is exceedingly rare [2]. This rarity underscores the importance of documenting such cases to enhance the understanding of RCC’s metastatic patterns and their clinical implications.

Acute abdomen, characterized by sudden and severe abdominal pain, is a medical emergency that demands prompt diagnosis and management. Its etiology is diverse, ranging from benign to life-threatening conditions, including complications from malignancies such as perforations caused by metastatic lesions. The management of acute abdomen in the context of underlying malignancy poses unique challenges, requiring a multidisciplinary approach for optimal patient care [3].

## 2. Case Presentation

The subject of this detailed case report is a 59-year-old male who had a clinically significant medical history, primarily notable for renal cell carcinoma. The patient had previously undergone a radical nephrectomy four years prior as part of his RCC treatment, at which point the disease was at Stage I with negative surgical margins. Since the surgery, he had been regularly monitored for signs of recurrence or metastasis through routine oncological follow-ups. His medical history was otherwise unremarkable, aside from being treated for arterial hypertension, with no reported gastrointestinal disorders or previous abdominal surgeries, making his current presentation particularly unusual and concerning.

### 2.1. Clinical Findings

The patient’s presentation to the emergency department was precipitated by the acute onset of severe abdominal pain, accompanied by nausea and vomiting. Upon initial evaluation, the patient appeared pallid and acutely unwell. Vital signs were notable for tachycardia and fever. Physical examination of the abdomen revealed pronounced tenderness and rigidity, predominantly in the upper quadrants, suggesting peritoneal irritation.

### 2.2. Diagnostic Assessment

Initial laboratory investigations revealed a significantly elevated white blood cell count (WBC 15.64 × 10^9^/L; reference value: 4.0–10.0 × 10^9^/L) and raised levels of C-reactive protein (CRP 208.1 mg/L; reference value: <5 mg/L), both markers suggestive of an acute inflammatory response. To further elucidate the cause of the patient’s acute abdominal presentation, an urgent abdominal computed tomography (CT) scan with intravenous contrast, specifically Ultravist^®^ 370 (Bayer, Leverkusen, Germany), was performed, and the late arterial phase was shown (Figure 1). At CT, a soft, circumferential thickening of the jejunal wall was observed, demonstrating intense post-contrast enhancement alongside infiltration of the mesenteric fat tissue anteriorly and dilated small bowel loops. Additionally, altered mesenteric lymph nodes exhibiting intense enhancement were noted. The described CT findings pertain specifically to the identified lesion in the jejunum, situated distally to the ligament of Treitz, and is characterized by a measurement of approximately 29 × 34 mm (Figure 1).

### 2.3. Therapeutic Intervention

In light of the clinical findings of peritoneal irritation, elevated inflammatory parameters, and CT signs of jejunal obstruction due to the presence of a tumor, the patient was expeditiously taken for laparotomy. Surgical exploration revealed signs of acute peritonitis with turbid exudate and a perforation in the proximal jejunum, attributable to a solitary tumorous lesion. A resection of approximately 15 cm of the small bowel encompassing the lesion was performed. Subsequently, the restoration of bowel continuity was achieved through a stapled side-to-side anastomosis, utilizing surgical staplers to reestablish the integrity of the intestinal tract.

### 2.4. Follow-Up and Outcomes

The patient’s postoperative hospital course was marked by complications, notably the onset of respiratory distress followed by the development of pneumonia, a common yet grave postoperative complication. These were classified as Clavien-Dindo Grade II complications due to the management requiring pharmacological treatment with antibiotics beyond the basic postoperative care. Despite aggressive antibiotic therapy and exhaustive supportive care, there was a progressive deterioration in the patient’s clinical status. This decline culminated in the development of sepsis, precipitating multi-organ failure. This sequence of events significantly highlighted the intricacies and challenges inherent in the patient’s case. Regrettably, the patient succumbed to these complications ten days subsequent to the surgical intervention.

### 2.5. Histopathological Evaluation of Surgical Specimen

Evaluation of the resected surgical specimen revealed a grey, lobulated mass measuring 60 × 40 × 30 mm with a perforation site. Histopathological examination of the mass demonstrated a solid construction composed of a syncytium of tumor cells. These cells had a polygonal appearance with clear cytoplasm and were moderately pleomorphic. The nuclei were round, vesicular, and exhibited moderate polymorphism. Tumor infiltration was observed permeating the entire thickness of the wall, with fields of necrosis also present. Signs of invasive growth into lymphatic and blood vessels were noted (Figure 2).

Immunohistochemical staining was positive for vimentin, AE1/AE3, and epithelial membrane antigen (EMA) (Figure 2). AE1/AE3 is a broad-spectrum cytokeratin marker, commonly used to identify epithelial cells in histopathological analysis. The staining pattern was indicative of epithelial origin, supporting the diagnosis of metastatic RCC. In contrast, the specimen was negative for CK7 and CD10. The absence of CK7 and CD10 staining further confirmed the nature of the tumor as metastatic clear cell RCC. These histopathological and immunohistochemical findings were consistent with a diagnosis of metastatic clear cell renal RCC, as illustrated in Figure 2 of the report.

## 3. Discussion

The case presented herein illustrates a rare and intricate clinical scenario involving the metastasis of renal cell carcinoma (RCC) to the small intestine, resulting in perforation and subsequent acute abdomen. The rarity of this clinical presentation is highlighted by the fact that gastrointestinal metastases from RCC are uncommon, and when they do occur, they seldom present as acute abdomen due to perforation.

### 3.1. General Characteristics of Renal Cell Carcinoma

Renal cell carcinoma is a significant contributor to adult malignancies, comprising about 2% of all such cancers and often presenting in an insidious manner. It stands as the seventh most common neoplasm in the developed world. RCC is the predominant form of malignant neoplasia in the kidney, accounting for 90% of all renal solid tumors. This cancer type originates from the renal cortex and specifically arises from the proximal tubular epithelium of the kidney. The clear cell subtype of RCC is the predominant form, making up about 90% of all RCC cases. This subtype is characterized by a highly variable clinical course, which presents a significant challenge in terms of management and treatment [4].

### 3.2. Epidemiological Trends of Renal Cell Carcinoma

In the realm of global health, the incidence of RCC has been witnessing a notable increase. Specifically, in the United Kingdom, there has been an increase of 47% in RCC cases over the past decade [5]. This trend is even more pronounced in the United States, where there has been a staggering 50% increase in RCC incidence over the last 30 years [6]. This uptick is partially attributed to the incidental discovery of tumors during imaging for unrelated conditions. Moreover, the surge in modifiable risk factors like smoking, obesity, hypertension, diabetes, diet, alcohol consumption, and occupational exposure, coupled with lifestyle changes toward Western habits, has further fueled this increase, especially in developed countries. This situation is complicated by the fact that RCC is known as the “great mimicker,” making it difficult to differentiate between benign and malignant lesions based solely on incidental imaging findings [7]. Such rising numbers are concerning, particularly considering that deaths worldwide from kidney cancer, predominantly driven by metastasis to other organs, exceed 100,000 annually [4]. This escalating trend underscores the growing health burden of RCC and highlights the critical need for enhanced detection, treatment, and understanding of this malignancy.

### 3.3. Risk Factors and Etiology of Renal Cell Carcinoma

The etiology of RCC remains not fully elucidated, although several risk factors have been recognized. Notable among these are advanced age, hypertension, nephrolithiasis, and polycystic kidney disease [4]. Additionally, hereditary conditions such as tuberous sclerosis, Von Hippel–Lindau syndrome, hereditary papillary renal cancer, hereditary leiomyoma, familial renal oncocytoma and hereditary renal cancers, and Birt–Hogg–Dubé syndrome have been identified as contributing factors to the increased risk of RCC [8]. Occupational exposure to specific hazards, including asbestos and herbicides, is also associated with a heightened risk of developing RCC [4]. Despite these known risk factors, the majority of RCC cases are sporadic and are often diagnosed incidentally, indicating a complex interplay of genetic, environmental, and lifestyle factors in the development of this malignancy [4].

### 3.4. Metastatic Behavior of Renal Cell Carcinoma

Renal cell carcinoma is known for its potential to be diagnosed at an advanced stage, with 30 to 70% of cases presenting with local infiltration or distant metastases at the time of diagnosis [1]. This high rate of late-stage discovery is largely due to the disease’s asymptomatic nature, which often remains unnoticed until incidental imaging for unrelated reasons reveals the malignancy. RCC can metastasize to almost any part of the body, with the most frequent sites of metastasis being the lungs (50–60% of patients with metastatic disease), bones (30–40%), liver (30–40%), and brain (5%) [9]. According to Ritchie and deKernion [10], RCC presents with metastatic disease in 23% of cases, and metastasis can occur within 5 years of nephrectomy in 25% of cases. Remarkably, there is no defined time limit to the metastatic activity of RCC, with late metastatic disease diagnosed even after a 5-year period in 10% of patients [11]. Metastasis can also occur post-curative resection with clear margins (R0) in approximately 40% of patients [11]. Often, more than one organ system is involved in the metastatic process [12]. RCC metastases have been found to present even after nephrectomy, with cases reported up to 17.5 years post-nephrectomy [13]. Metachronous metastatic disease may develop in up to 50% of patients who have undergone a presumably curative radical nephrectomy [14].

The natural history of RCC can be highly unpredictable, posing significant diagnostic and management challenges. Due to this unpredictable nature, RCC often presents at advanced stages with metastases to a wide variety of organs [15]. Despite this, synchronous or late metastasis to the small bowel, as observed in this case, has been reported only in a limited number of cases. The jejunum is an even rarer site for metastasis, making this case a valuable contribution to the existing body of literature by documenting an atypical manifestation of RCC metastasis.

### 3.5. Metastatic Patterns and Rarity of Small Intestine Involvement in RCC

Metastases of renal cell carcinoma in the small intestine are considered a rare occurrence [16]. In the context of RCC, solitary metastases are infrequent, occurring in less than 2% of patients [17]. Notably, this case represents the first documented instance of solitary metastasis to the small intestine manifesting as perforation and presenting with symptoms of acute abdomen.

RCC is known for its tendency to metastasize through various routes. The primary mechanisms of spread include hematogenous pathways as well as, less commonly, lymphatic channels. Additionally, the spread of RCC can occur through transcoelomic means or by direct invasion [15]. The propensity for RCC to disseminate via these multiple pathways underscores the complexity of its metastatic behavior and the challenges it poses in both diagnosis and management.

### 3.6. Clinical Presentation of Gastrointestinal Metastases in RCC

The diagnosis of gastrointestinal (GI) metastases originating from RCC is often challenging and typically delayed, as these metastatic lesions are usually identified following the onset of clinical symptoms [16]. The predominant clinical presentation of GI metastases from RCC is gastrointestinal bleeding, which occurs in about 67% of cases due to the invasion of intestinal vessels by the metastatic lesion [18]. Other manifestations can include intestinal obstruction or symptoms related to the mass effect of the metastasis. There is a crucial recommendation for clinicians to consider metastatic disease as a source of bleeding in patients with a known history of RCC who present with GI bleeding [18]. This approach is vital for early detection and appropriate management of these metastases.

Interestingly, while obstruction, anemia, pain, nausea, vomiting, and weight loss are common symptoms associated with these metastases, the current case stands out as it represents a rare instance where the metastasis of RCC to the small intestine has manifested as a perforation of the small bowel. This is a significant deviation from the typical presentations and has not been previously documented in the literature.

### 3.7. Literature Review and Demographics of Small Bowel Metastases in RCC

A comprehensive literature search conducted using PubMed with the criteria “metastatic AND RCC OR renal cell carcinoma AND small bowel” identified 60 published case reports or series featuring symptomatic, metastatic RCC to the small bowel. Of these cases, 11 were solitary metastases similar to the case observed in our patient (Table 1).

As delineated in Table 1, the foundational case of metastatic renal cell carcinomato the small intestine, first documented by Starr A et al. [19] in 1952, laid the groundwork for understanding this clinical phenomenon. It is notable that a significant number of cases have been reported in the last two decades, suggesting an increased awareness and documentation of this condition. A notable trend in recent years is the potential link between the rise in reported cases and the prolonged survival of RCC patients, attributed to advancements in targeted chemotherapy [27]. This extended survival could allow for more opportunities for primary tumors to metastasize to the small bowel and manifest symptomatically.

Analysis of Table 1 reveals that metastatic RCC to the small intestine predominantly affects males and spans a broad age spectrum, with an average onset age of about 61 years. The time between nephrectomy and metastasis averages approximately 8 years, with the longest interval recorded at 20 years post-surgery.

In the cases studied, solitary metastasis to the small intestine commonly presented as gastrointestinal bleeding in half of the instances. Obstruction of the small bowel, often due to intussusception, was noted in five cases. This is frequently attributed to RCC metastases forming pedunculated tumors that act as lead points for intussusception. Uniquely, our case represents a solitary RCC metastasis presenting as a perforation, setting it apart from other cases. Surgical management varied but typically involved resection and anastomosis, with patient outcomes ranging from successful recoveries to fatalities. These findings highlight the complex and variable prognosis associated with metastatic RCC.

### 3.8. Diagnostic Approach to Intestinal Metastasis of RCC

Diagnosing metastatic RCC in the small bowel presents significant challenges [28]. In patients who have undergone nephrectomy for RCC and subsequently present with gastrointestinal symptoms, a comprehensive diagnostic approach is essential [28]. This should include both endoscopic and radiologic evaluations to accurately assess the extent of metastatic disease. Additionally, it is important to recognize that RCC metastases to the small intestine can manifest as bleeding or signs of small bowel obstruction.

The small bowel poses particular difficulties for standard endoscopic examination and is often not adequately visualized in barium studies [29]. The optimal imaging strategy for assessing the small bowel remains a subject of debate [30]. Jejunal metastases, which can manifest many years post-initial RCC diagnosis, may present as bleeding and necessitate extensive endoscopic explorations. These explorations include enteroscopy, videocapsule endoscopy, or exploratory surgery combined with enteroscopy, owing to the challenges associated with their detection. [31]. Enteroscopy, performed via both oral and anal approaches, is recommended and should not be confined to the upper digestive tract [32]. CT/MR enterography is the preferred modality for the diagnosis and staging of these conditions. PET-CT can also be useful for initial diagnosis and staging, although it is not routinely recommended.

When diagnosing metastatic RCC to the small bowel, CT imaging plays a crucial role in identifying signs of bowel obstruction, a potential complication of this condition. CT scans can reveal partial or complete obstructions in the small bowel, showcasing transition points marked by solid enhancing masses indicative of the presence of metastatic disease [33]. Moreover, abdominal CT scans are particularly effective in confirming intussusception, a condition that can be caused by metastatic spread to the small intestine from RCC. The diagnostic accuracy and sensitivity of abdominal CT in identifying intussusception, including signs of bowel wall edema and the appearance of a lead mass, can reach up to 100%, making it an invaluable tool in the diagnostic process [14,16]. Despite its utility, small bowel video capsule endoscopy has limitations, including the inability to obtain tissue samples and its restricted use in cases of small bowel obstruction, with a reported false-negative rate of up to 18.9%. Capsule endoscopy and push or double balloon enteroscopy (DBE) may be necessary in patients with gastrointestinal symptoms and a history of RCC [25].

The combination of videocapsule endoscopy for screening and enteroscopy for obtaining histological samples is emerging as an effective approach to detecting small intestine metastases, particularly in the context of digestive bleeding with normal results from upper and lower endoscopic explorations [31]. In cases of massive digestive hemorrhage, the diagnosis can be made through angiography or during surgery [31].

### 3.9. Treatment Strategies for Metastatic RCC

The treatment landscape for metastatic renal cell carcinoma has undergone significant changes, especially in the context of metastatic disease management. Treatment options for RCC metastases encompass surgical intervention as well as various interventional therapies, which have demonstrated efficacy in enhancing patient survival [27]. Specifically, the use of the tyrosine kinase inhibitor Sunitinib has been shown to provide survival benefits for patients with RCC metastasis [34]. Additionally, newer therapies targeting the vascular-endothelial growth factor receptor (VEGFR) and the mTOR-signaling pathway have shown promising results in the treatment of RCC metastases [35].

Over the past 15 years, the management of metastatic RCC (mRCC) has evolved considerably with the introduction of tyrosine kinase inhibitors (TKIs) [36]. More recently, the advent of immunotherapy, particularly immune checkpoint inhibitors, has further revolutionized treatment. The use of these immunotherapies, either alone or in combination with TKIs, has significantly extended the lifespan of patients with metastatic clear cell renal cell carcinoma (mccRCC) [37].

#### 3.9.1. Surgical Management of RCC Metastases

With advancements in targeted therapies in recent years, the prognosis for patients with metastatic RCC has significantly improved [27]. However, in scenarios where targeted therapies do not yield dramatic results, surgical excision of isolated metastases continues to be a vital component in the treatment strategy for metastatic RCC [38]. Complete resection of the metastatic lesions is considered the most effective treatment for RCC metastases, and surgery remains the treatment of choice for localized metastatic RCC [39].

The primary therapeutic goal is to achieve complete metastasectomy whenever it is surgically feasible. Undertaking any type of metastasectomy can enhance patient survival [39]. In our case, surgical intervention was absolutely indicated due to the development of an acute abdomen resulting from the perforation of an RCC metastasis. The procedure involved the resection of the small intestine containing the metastasis and the establishment of digestive tract continuity using a side-to-side anastomosis.

Beyond curative surgeries, there are situations where small intestinal metastatic lesions are unresectable and accompanied by acute intestinal obstruction or bleeding. In these cases, palliative interventions such as enterostomy, bypass surgery, or urgent selective arterial embolization can offer symptomatic relief and potential benefits to the patients [38]. These approaches underscore the importance of individualized treatment plans, especially in complex cases of metastatic RCC where surgical options are weighed against the patient’s overall condition and prognosis.

#### 3.9.2. Systemic Therapy in the Treatment of Metastatic RCC

The systemic treatment of metastatic RCC has advanced significantly with the introduction of targeted therapies and immunotherapy, leading to improved prognosis for patients [27]. RCC is recognized as a metabolic disease involving abnormal alterations in oxygen-sensing metabolic pathways, leading to the upregulation of hypoxia-inducible factor (HIF) pathways and related genes such as PDGF, VEGF, and epidermal growth factor [40]. These pathways have been targeted using tyrosine kinase inhibitors (TKIs) like sunitinib and sorafenib, improving overall survival [41]. Sunitinib, a TKI targeting angiogenic receptors commonly upregulated in metastatic RCC, has become a first-line therapy for these patients. It targets PDGF receptors, c-KIT, FLT3, VEGF receptor-1, and VEGF receptor-2, showing significant enhancement in overall survival and clinical outcomes [42].

The mechanistic targets of rapamycin (mTOR) inhibitors, focusing on the frequently activated mTOR signaling in RCC, have been critical despite their varying success. They target elements of the mTOR signaling pathway leading to cancer cell proliferation, survival, and angiogenesis [43]. Furthermore, the use of VEGF antibodies, specifically bevacizumab, has been explored in RCC. Bevacizumab works by inhibiting angiogenesis, a critical process in tumor growth and metastasis [44]. The incorporation of these therapies, along with immunotherapy and TKIs, represents the evolving landscape of metastatic RCC treatment, where combination and targeted therapies are becoming increasingly significant.

Additionally, other forms of immunotherapy, particularly the combination of nivolumab (a PD-1 inhibitor) and ipilimumab (a CTLA-4 inhibitor), have shown superior outcomes in overall survival and response rates compared to sunitinib in patients with intermediate or poor risk metastatic RCC [45]. This regimen has become the new standard-of-care first-line treatment for these patients, though it is important to consider potential immune-related adverse effects. Moreover, the use of immune checkpoint inhibitors in combination with anti-VEGF targeted agents has shown promising results in trials like JAVELIN Renal 101 and Keynote-426, offering higher response rates and progression-free survival compared to traditional treatments [46]. These combinations have received FDA approval for first-line treatment of metastatic RCC, marking a significant shift in the treatment paradigm [47].

### 3.10. Surveillance Strategies in Metastatic RCC Post-Treatment

The follow-up intensity and duration for patients treated for renal cell carcinoma remain a subject of debate. Various surveillance programs are based on different risk scores to evaluate recurrence risk. The 2023 update of the European Association of Urology guidelines emphasizes the importance of tailoring surveillance algorithms to each patient’s risk profile and treatment efficacy [48]. This individualized approach is essential for optimizing follow-up strategies and has been shown to offer survival benefits.

Both the National Comprehensive Cancer Network (NCCN) and the American Urology Association (AUA) recommend routine postoperative surveillance for the first five years. However, due to the potential for late RCC recurrence, extending surveillance beyond this period may be beneficial, as longer follow-up periods have been associated with reduced recurrence rates [49,50]. The need for prolonged surveillance highlights the complexity of managing RCC and supports a multidisciplinary approach, considering factors such as the type of surgery, histological RCC type, and patient-specific factors like comorbidities and genetic profiles.

As elaborated in Table 2, the complexity of RCC metastasis to the small bowel is underscored by several key findings and considerations. This table captures the essence of the rarity and unpredictability of such metastasis, its delayed presentation, and the diverse clinical manifestations it can cause. The challenges in diagnosis, underscored by its atypical presentations, call for a meticulous and multi-faceted approach in both detection and treatment.

## 4. Conclusions

In conclusion, this comprehensive case study and literature review elucidate the rare and complex phenomenon of small bowel perforation due to metastatic renal cell carcinoma. This case underscores the unpredictable nature of RCC metastases, particularly to the small intestine, and highlights the importance of ongoing vigilance and surveillance in patients with a history of RCC. The variety of clinical presentations, from gastrointestinal bleeding to bowel obstruction, and in this unique case, perforation leading to acute abdomen, emphasizes the need for a multidisciplinary and individualized approach to diagnosis and management. The evolving landscape of RCC treatment, with advancements in surgical techniques and systemic therapies, offers new hope but also presents challenges in managing this aggressive cancer type.

## Figures and Tables

**Figure 1 diagnostics-14-00761-f001:**
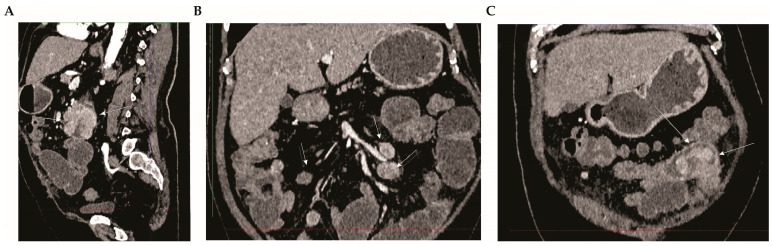
Radiological Identification of Metastatic Renal Cell Carcinoma (RCC) in the Small Intestine. Panel (**A**)—Sagittal view: White arrows indicate a soft, circumferential thickening of the jejunum wall, which exhibits intense post-contrast enhancement. There are also signs of mesenteric fat infiltration and dilated small bowel loops. Panel (**B**)—Coronal view: Arrows highlight pathologically altered mesenteric lymph nodes, characterized by intense enhancement. Panel (**C**)—Sagittal view: A soft tissue change in the jejunum is evident (white arrows), showing intense post-contrast enhancement, alongside dilated small bowel loops. The CT scan was performed on a Siemens SOMATOM go. Top scanner using Ultravist 370 as the contrast. The abdomen-pelvis protocol included the patient in a supine position, centered within the gantry, arms elevated, with a craniocaudal scan direction, and a scan thickness of 1 mm. The reconstruction algorithm included soft tissue and bone kernel, with no oral contrast administered. The contrast volume was 100 mL (2.5 mL/kg), with no saline chaser, using bolus tracking of the abdominal aorta. Multiplanar reconstruction images were performed in axial, sagittal, and coronal planes.

**Figure 2 diagnostics-14-00761-f002:**
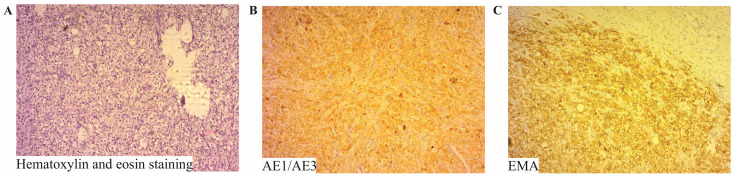
Detailed Histopathological Characterization of Tumor Tissue. Panel (**A**) presents the histopathological examination of the tumor mass, utilizing hematoxylin and eosin staining. This analysis reveals a solid construction of tumor cells forming a syncytium. These cells are polygonal with clear cytoplasm and exhibit moderate pleomorphism. Their nuclei are round, vesicular, and demonstrate moderate polymorphism, indicative of their neoplastic nature. In Panel (**B**,**C**), the immunohistochemical analysis further clarifies the cellular characteristics, showing a strong positivity for AE1/AE3 (**B**) and epithelial membrane antigen (EMA) (**C**) in the tumor cells. This combination of staining techniques, magnified at ×40, provides a comprehensive view of the tumor’s cellular architecture and molecular profile, essential for accurate diagnosis.

**Table 1 diagnostics-14-00761-t001:** Comprehensive Overview of Solitary RCC Metastasis to the Small Intestine: Case Summaries.

Author	Age/ Gender	Manifestation	RCC Diagnosis History	Histopathology (Primary Tumor)	Metastasis Location	Metastatic RCC Treatment	Metastatic RCC Histopathology	Postoperative Therapy
Starr A et al. (1952) [19]	52/F	Obscure GI bleeding, anemia	20 years prior, nephrectomy	RCC isolated in renal parenchyma	Middle part of jejunum	Jejunum segment resection, anastomosis	Metastatic clear cell RCC	N/A
Toh SK et al. (1996) [20]	59/F	Colicky abdominal pains, indigestion, anorexia, weight loss	10 years prior, nephrectomy with splenectomy	Stage 1 RCC	Fourth part of duodenum	Duodenotomy, peduncular mass excision	Metastatic RCC	None
Venugopal A et al. (2007) [21]	54/M	Melena, intussusception	6 years prior, nephrectomy	RCC isolated to renal parenchyma	Mid ileum	Ileum segmental resection, end-to-end anastomosis	Metastatic RCC	N/A
Bahli ZM et al. (2007) [22]	65/F	Small bowel obstruction	1 year prior, nephrectomy, adrenalectomy	T2G4 RCC and pheochromocytoma	Region of jejunum	Small bowel resection, end-to-end anastomosis	Metastatic RCC	N/A
Vazquez C et al. (2011) [23]	68/M	Obscure occult GI bleeding	1 year prior, radical nephrectomy	Clear-cell RCC (pT2N × M0EII)	Proximal jejunum	Enteroscopy, tumor excision	Clear cell renal metastasis	N/A
Geramizadeh B et al. (2015) [24]	61/M	GI bleeding	16 years prior, nephrectomy	Clear cell RCC	Second part of duodenum	Whipple’s operation, pancreatoduodenal mass resection	Metastatic RCC	N/A
Ismail I et al. (2015) [14]	66/M	Vomiting, abdominal pain	19 years prior, radical nephrectomy	Localized clear-cell type RCC (T1aN0M0)	Jejunum	Wide margin resection, end-to-end anastomosis	Polypoid metastatic RCC	None
Gorski RL et al. (2015) [25]	82/M	Black stools	6 years prior, nephrectomy	RCC with vascular invasion and lymph node metastasis	Proximal jejunum	None	N/A	Declined treatment
Mueller JL et al. (2018) [18]	63/M	Bright red blood per rectum	3 years prior, partial nephrectomy	pT1a clear cell RCC (Fuhrman grade 3/4)	Terminal ileum	Ileum segment resection, side-to-side anastomosis	Metastatic clear cell RCC	None
Kim D et al. (2023) [2]	60s/M	Constipation, nausea, vomiting, small bowel obstruction	6 years prior, nephrectomy, pembrolizumab	Multifocal clear RCC (Fuhrman grade 3, T3a NX)	Distal jejunum, proximal ileum	Affected small bowel segment removal, side-to-side anastomosis	Metastatic RCC with sarcomatoid feature	N/A
Leal PV et al. (2023) [26]	50/F	Intussusception	N/A, previous nephrectomy	Stage 2 clear cell RCC (pT2cN0cM0)	Proximal jejunal intussusception	Jejunal segment resection	Clear cell renal metastasis	N/A
Current case (2024)	59/M	Perforation, acute abdomen	4 years prior, radical nephrectomy	RCC confined to kidney	Proximal part of jejunum	Jejunal segment resection, side-to-side stapled anastomosis	RCC metastasis	Complicated post-op course

Note: RCC, Renal Cell Carcinoma; GI, Gastrointestinal; PN, Partial Nephrectomy; OS, Overall Survival; N/A, Not Available.

**Table 2 diagnostics-14-00761-t002:** Key Insights on Renal Cell Carcinoma Metastasis to the Small Bowel: Challenges and Considerations.

Key Finding or Consideration	Description
Rarity of Occurrence	RCC metastasis to the small bowel is exceptionally rare, making each case a valuable contribution to medical knowledge.
Delayed Presentation	Metastasis can occur several years post-nephrectomy, necessitating long-term vigilance and follow-up.
Varied Clinical Presentations	Manifestations range from gastrointestinal bleeding and bowel obstruction to unique cases like perforation leading to acute abdomen.
Diagnostic Challenges	Due to its rarity, RCC metastasis to the small bowel can be difficult to diagnose, often requiring extensive investigation.
Importance of Histopathology	Detailed histopathological examination is crucial for confirming the diagnosis of RCC metastasis.
Multidisciplinary Treatment Approach	Effective management often involves a combination of surgical intervention and systemic therapies.
Need for Individualized Patient Care	Treatment and follow-up strategies should be tailored to each patient’s unique clinical scenario.

## Data Availability

This article, which encompasses a case report and review, does not include any new primary data for dissemination. The information discussed is sourced from previously published research and the patient’s medical records.

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
