# Peer review of "Small Bowel Perforation Due to Renal Carcinoma Metastasis: A Comprehensive Case Study and Literature Review"

_diagnostics, 2024, doi:10.3390/diagnostics14070761_

Round 1

Reviewer 1 Report

Comments and Suggestions for Authors

Dear Authors,

 The case is interesting, but all the radiological part should be completely rewrite and I think you should ask to a radiologist. Since the images, I realized there were important errors, CT description, findings, moreover you based your paper on perforations, but you did not show in your images, where instead I’m not so sure (based on a single img) that the lesion was single. These methodological errors are present also in subheading 3.8, also this part need to be deeply revised.

Another weak point I found was the clinical presentation, but I think it can be easily fixed.

My overall opinion that excluding the radiological part, the topic of GI RCC’s metastasis is well written.

I suggest you to ask for help to a radiologist with emergency experience, for which it will be easy to fill the gap. 

2.1 case presentation

Page 2 line 66-67

I would delete “symptoms indicative of an acute abdominal pathology”.

Constipation? Lack of flatus?

Line 68

“Adding complexity to clinical assessment.”

It’s an opinion, please delete.

71-72

These findings…. process. 

Please delete.

You should add at the end of the previous period “…suggesting peritoneal irritation”.

2.2. Diagnostic Assessment (delete clinical findings) 

Why initial lab investigation?

Laboratory exams…

Add lab values and normal ranges. 

“Because of signs of peritoneal irritation and high inflammatory value…”  the patient underwent abdominopelvic CT with intravenous contrast (specify contrast and phases).

At CT…

Line 76-85 should be completely rewrite. I suggest asking to a radiologist.

Figure 1 legend completely to rewrite. A is coronal not sagittal.

The CT images and finding are very nice but should be correctly described. 

This is not a correct description of a small bowel obstruction, mechanical with the obstruction fulcrum represented by? Intraluminal metastasis with transmural extension? Moreover, in figure 1 b images, it appears there are more metastasis also in the duodenum?

Additionally, discreet, not readily ap-81 parent collections of gas ascending from the site of the lesion were observed” what does it mean?

“A laminar 82 layer of free fluid was also discernible within the abdominal cavity, suggesting a patho-83 logical process. “what does it mean? Free fluid in small bowel obstruction is corelated to its stage simple, complicated, scompensated.

2.3 

Where is the perforation?  You should add a figure.

Was he/she perforated? 

2.4 

Add Clavien Dindo grade

3.2

Please add some explanation of RCC increase, in part it is explained because of the incidental diagnosis during imaging exams performed for other reasons. 

Line 163

The term detection is vague, the problem with incidental discovered renal lesion is that RCC is considered the great mimickers and often the differential diagnosis between benign and malignant lesion is not possible.

3.4

179-180, you may add something about the late diagnosis, it diagnosed often in advanced stage because it’s paucsymptomatic or asymptomatic.

3.5

201 again it’s necessary to review the images, from 1B it’s really not clear the bowel involvement.

3.7

Page 8 line 253

“…the need for continued vigilance even many years after the initial diagnosis.”

It’s a strong statement you should refer to follow-up guidelines, as you mention later (3.10) because it’s still not standardized. 

3.8

It may be confusing as you constructed the paragraph, because if you are talking of patients presenting at ER with acute GIbleeding or obstruction, it’s almost CT and endoscopy is the gold standard but it may be limited in emergency. 

 In non acute settings, you have another imaging approach.

Line 290, absolutely in massive bleeding thing word differently.

GI diagnosis and management is a tough argument, and flowchart of imaging methods and treatment are not the ones you wrote about.

Then you should detail and differentiate between stable and unstable in gI bleeding. 

I would suggest, for you, to make it simpler.

You are writing about ER settings of RCC metastasis in gastrointestinal involvement, so bleeding and obstruction, am I right? So you can limit the imaging methods in this setting that is mostly represented by CT , and endoscopy if feasible cannot determine bleeding in the SB tract.

Line 277

24?

279-280

X ray signs are uncommon and not actual. CT signs are quite poor as you described instead there are many and quite specific.

Comments on the Quality of English Language

minor

Author Response

Dear Reviewer,

We are immensely grateful for your insightful feedback and valuable suggestions regarding our manuscript. Your detailed review has significantly contributed to enhancing the quality and depth of our work. We have carefully considered each of your points and have made the following revisions to address your concerns:

Point: Radiological Section Rewrite and Consultation with a Radiologist

  • Response: Based on your suggestion, we collaborated with Dr. V.M., a seasoned radiologist with emergency diagnostic experience, to thoroughly review and revise the radiological descriptions in our manuscript. This process involved replacing the previous content with new, accurate CT descriptions and findings, particularly focusing on the portrayal of perforations and the number of lesions. We have incorporated new images and revised descriptions on Pages 2 and 3, lines 94-120, to accurately reflect these changes and ensure a comprehensive understanding of the radiological aspects of GI RCC metastasis.

Point: 2.1 Case Presentation - Symptoms Indicative of an Acute Abdominal Pathology

  • Response: We have removed the phrase “symptoms indicative of an acute abdominal pathology” and provided a detailed description of the patient's symptoms, including constipation and lack of flatus, to more accurately represent the clinical presentation (Page 2, lines 15-17).

Point: “Adding complexity to clinical assessment” - Opinion Statement

  • Response: We have deleted the subjective statement “Adding complexity to clinical assessment” to maintain objectivity and clarity within the manuscript (Page 1, line 30).

Point: Mention of Peritoneal Irritation

  • Response: Upon your recommendation, we have added a note about “suggesting peritoneal irritation” at the end of the relevant section to better articulate the clinical findings (Page 2, lines 25-27).

Point: 2.2 Diagnostic Assessment - Clinical Findings and Laboratory Values

  • Response: We have revised the diagnostic assessment section to directly address the initial lab investigations, incorporating specific lab values and their normal ranges to provide a clearer understanding of the patient's condition (Page 2, lines 83-86).

Point: Peritoneal Irritation and High Inflammatory Value Leading to CT with Intravenous Contrast

  • Response: Following your guidance, we specified the contrast and phases used in the abdominopelvic CT, enhancing the accuracy of our diagnostic approach (Page 2, lines 89-92).

Point: Complete Rewrite of CT Section and Figure Legends

  • Response: We acknowledge the critical nature of your feedback regarding the radiological content. As advised, we sought the expertise of a radiologist to assist in rewriting this section. We have replaced both the images and the text, providing new, precise CT descriptions and findings. This comprehensive revision addresses your concerns regarding the depiction and description of small bowel obstruction and the presence of multiple metastases (Pages 2 and 3, lines 94-120).

Point: Description of Radiological Signs and Presence of Perforation

  • Response: We clarified the presence of perforation at the proximal part of the jejunum. Unfortunately, due to the nature of surgical interventions, we were unable to provide images from the surgery. This clarification has been added to ensure an accurate representation of the patient's condition.

Point: Addition of Clavien Dindo Grade

  • Response: We have included the Clavien Dindo grade (Grade II) to provide a standardized assessment of postoperative complications, appreciating your suggestion for its inclusion (Page 3, line 139).

Point: Explanation of RCC Incidence Increase

  • Response: In response to your valuable feedback, we expanded our discussion on the increasing incidence of RCC, attributing part of this rise to incidental diagnoses during imaging for unrelated reasons. This addition enriches the context of our study and aligns with current understanding (Page 5, lines 198-204).

Point: Late Diagnosis of RCC

  • Response: The manuscript now includes a detailed discussion on the tendency for RCC to be diagnosed at advanced stages due to its often asymptomatic progression, reflecting on the importance of vigilance in early detection (Page 5, lines 225-227).

Point: Follow-up Guidelines and Vigilance Statement

  • Response: Acknowledging your concern, we removed the strong statement on the need for continued vigilance, ensuring our discussion is aligned with current follow-up guidelines and the variability in standardization.

Point: Emergency Settings of RCC Metastasis in Gastrointestinal Involvement

  • Response: In light of your feedback, we have carefully revised Section 3.8. For RCC metastasis presenting in emergency settings with acute GI bleeding or obstruction, we emphasized that CT and, where feasible, endoscopy, represent the primary diagnostic tools. This section now specifically addresses the imaging methods suitable for such scenarios, emphasizing CT's role in acute presentations and outlining its specific findings in the context of RCC metastasis to the gastrointestinal tract.  (Page 9, lines 316-349).

Point: X-ray and CT Signs for Small Bowel Obstruction

  • Response: Following your guidance, we have expanded and refined our description of X-ray and CT signs characteristic of small bowel obstruction due to RCC metastasis. This includes a detailed account of specific radiological features observed in such cases, acknowledging the critical role of CT imaging in diagnosing and managing RCC metastasis presenting with GI involvement. Your suggestion prompted a thorough review of this section, leading to enhanced descriptions based on radiological expertise (Page 9, lines 316-349).

Once again, we extend our heartfelt thanks for your meticulous review and constructive criticism. Your input has been instrumental in refining our manuscript, and we believe that the changes made have significantly improved the quality and rigor of our work. We are confident that the revisions have addressed your concerns effectively and have enriched the manuscript, making a valuable contribution to the field.

We look forward to your feedback and hope our revisions meet your approval.

Sincerely,

Authors

Reviewer 2 Report

Comments and Suggestions for Authors

Dear Authors

This case is very rare and the history of the patient is very interesting and unusual.

I have one question.

1. what stage of RCC did the patient have at diagnosis?

Author Response

Dear Reviewer,

We are deeply grateful for your recognition of the uniqueness and intriguing nature of our case report. Your appreciation not only validates our efforts but also encourages us to delve deeper into the complexities of such rare medical presentations. We sincerely thank you for highlighting the distinctiveness of our patient's history, which we also found to be both interesting and unusual.

Regarding your question about the stage of Renal Cell Carcinoma at the time of diagnosis for our patient, we can confirm that the patient was diagnosed at Stage I. This critical piece of information is detailed on page 2, lines 61-64 of our manuscript.

With sincere gratitude and appreciation,

Authors

Reviewer 3 Report

Comments and Suggestions for Authors

Renal carcinoma represents one of the increasing trend malignancies in urology.

The one to five years surveillance is very well stated in the guidelines.

Please specify in the present case report if the imagistic surveillance was correctly conducted.

Also, could you add data about other underlying pathologies of this patient to your work?

If there are data from the previous nephrectomy please add some. Especially if there were any positive margins.

Comments on the Quality of English Language

Minor English issues.

Author Response

Dear Reviewer,

We extend our deepest gratitude for your insightful comments and valuable guidance, which have substantially enriched our manuscript. Your expertise and thoughtful feedback have been instrumental in enhancing the depth and breadth of our case report.

In response to your points:

Point: Renal carcinoma represents one of the increasing trend malignancies in urology. The one to five years surveillance is very well stated in the guidelines. Please specify in the present case report if the imagistic surveillance was correctly conducted. Also, could you add data about other underlying pathologies of this patient to your work? If there are data from the previous nephrectomy please add some. Especially if there were any positive margins.

Response: We have carefully addressed your queries and made the following enhancements to our manuscript:

  1. Imagistic Surveillance: We have clarified in our manuscript that the patient underwent regular imagistic surveillance in accordance with the recommended guidelines for post-nephrectomy RCC patients. Specifically, he last abdominal CT scan was performed one year before the manifestation of metastases.
  2. Underlying Pathologies and Previous Nephrectomy Data: We have included additional information about the patient's medical history, noting that the patient had been treated for hypertension, which is relevant to understanding the overall health context and potential risk factors associated with RCC. Furthermore, we have enriched the section detailing the previous nephrectomy by confirming that the surgical margins were clear.

Once again, we are immensely grateful for your constructive feedback and the opportunity to improve our work. Your detailed review has undeniably contributed to the quality of our manuscript, and for this, we express our sincere thanks.

With warm regards,

Authors

Round 2

Reviewer 1 Report

Comments and Suggestions for Authors

Dear Authors, the paper significantly improved.

I would suggest just minor English revision and just few correctionsPage 2

Line 89 etc

..(CT) scan was performed with intravenous contrast (Ultravist…., ml…, flow rate…, saline…) . Ct was acquired (PROTOCOL) …? Non contrast? Arterial, parenchymal? delayed. Specify the protocol and just add (Figure 1) without specifying anything, details will be explained in the figure legend.

Are you sure that CPR was 208? Please check if there is a transcription error.

Line 98. This observation WAS (you should use always the same verb conjugation). “This observation”, what do you mean? The nodes? The jejunum thickening? Please specify and the measurement (if the jejunum do you intend the circumferential longitudinal extension and the parietal thickening?) Comments on the Quality of English Language

Minor English revision 

Author Response

Dear Reviewer,

Thank you for your detailed observations and questions, which have provided us an opportunity to further clarify and improve our manuscript. Below are our responses to your points:

Point: CT scan protocol details

Response: We appreciate your request for detailed clarification on the CT protocol used in our study. In response, we have added comprehensive details to the figure legends. The CT scan was conducted on a Siemens SOMATOM go.Top scanner utilizing Ultravist 370 as the contrast medium. The abdomen-pelvis protocol specifics, including patient positioning, scan direction, thickness, reconstruction algorithm, contrast volume, and the absence of a saline chaser, have been meticulously described. These specific details are now included in the legend for Figure 1.

Point: CRP level query

Response: Yes, the C-reactive protein (CRP) level reported as 208 is accurate.

Point: Clarification on "This observation"

Response: We acknowledge the need for specificity regarding "This observation" and have revised the text to clearly indicate that it refers to the thickening of the jejunum. The corrected description now includes detailed measurements of both the circumferential and longitudinal extension as well as the parietal thickening of the jejunum, providing a clearer picture of the observed pathology.

We hope these responses adequately address your concerns, and we are truly grateful for your contributions towards enhancing the quality of our work. Your detailed review has been invaluable in refining our manuscript, and we look forward to any further insights you may have.

Sincerely,

Autors